# Challenges of CRISPR/Cas-Based Cell Therapy for Type 1 Diabetes: How Not to Engineer a “Trojan Horse”

**DOI:** 10.3390/ijms242417320

**Published:** 2023-12-10

**Authors:** Dmitry S. Karpov, Anastasiia O. Sosnovtseva, Svetlana V. Pylina, Asya N. Bastrich, Darya A. Petrova, Maxim A. Kovalev, Anastasija I. Shuvalova, Anna K. Eremkina, Natalia G. Mokrysheva

**Affiliations:** 1Engelhardt Institute of Molecular Biology, Russian Academy of Sciences, 119991 Moscow, Russia; aleom@yandex.ru (D.S.K.); sollomia@yandex.ru (A.O.S.); kovalev_maksim_2002@mail.ru (M.A.K.); anastasiashuvalova777@gmail.com (A.I.S.); 2Endocrinology Research Centre, 115478 Moscow, Russia; pylina@icloud.com (S.V.P.); asjaha@gmail.com (A.N.B.); darya0813@mail.ru (D.A.P.); eremkina.anna@endocrincentr.ru (A.K.E.)

**Keywords:** type 1 diabetes, β-cells, CRISPR/Cas, iPSC, cell therapy

## Abstract

Type 1 diabetes mellitus (T1D) is an autoimmune disease caused by the destruction of insulin-producing β-cells in the pancreas by cytotoxic T-cells. To date, there are no drugs that can prevent the development of T1D. Insulin replacement therapy is the standard care for patients with T1D. This treatment is life-saving, but is expensive, can lead to acute and long-term complications, and results in reduced overall life expectancy. This has stimulated the research and development of alternative treatments for T1D. In this review, we consider potential therapies for T1D using cellular regenerative medicine approaches with a focus on CRISPR/Cas-engineered cellular products. However, CRISPR/Cas as a genome editing tool has several drawbacks that should be considered for safe and efficient cell engineering. In addition, cellular engineering approaches themselves pose a hidden threat. The purpose of this review is to critically discuss novel strategies for the treatment of T1D using genome editing technology. A well-designed approach to β-cell derivation using CRISPR/Cas-based genome editing technology will significantly reduce the risk of incorrectly engineered cell products that could behave as a “Trojan horse”.

## 1. Introduction

Type 1 diabetes mellitus (T1D) is a chronic disease caused by the destruction of insulin-producing pancreatic β-cells by cytotoxic immune cells [1]. To date, there are no drugs or vaccines that can prevent the development of T1D. Due to the ineffectiveness of early screening methods, the diagnosis of the disease is not confirmed until the late stages, when approximately 90% of β-cells have been lost. It is estimated that 8.42 million people were living with T1D in 2021, and this number is expected to increase to 13.5–17.4 million by 2040 [2]. The standard treatment for patients with T1D is hormone replacement therapy. This treatment is life-saving but it is costly, has complications, can lead to acute and chronic complications, and ultimately results in reduced overall life expectancy [3]. Therefore, there is an intense search for alternative ways to treat T1D more effectively and safely.

Regenerative medicine offers a promising way to treat T1D using cell transplantation technologies. Patient-specific cell products derived from autologous induced pluripotent stem cells (iPSCs) and other cell types are an ideal option for transplantation in terms of avoiding an immune response. However, allografts are preferable to autografts for a number of reasons. This is primarily due to the high relative cost of obtaining patient-specific cell products. In addition, the use of cell products in medicine implies the collection of thorough preclinical data on each individual cell line, which is associated with significant expenditure of material resources and time. Therefore, the use of allogeneic cell products seems reasonable. At the same time, there is a pressing problem of the rejection of transplanted cells. Currently, there are two potential solutions: the creation of biobanks of iPSC lines for selection of cells with the best immunocompatibility and creation of immunoprivileged iPSC lines. Biobanking seems to be a rational approach for countries with low genetic diversity, such as Japan [4,5]. However, in the case of large heterogeneous countries, this approach is not suitable or has relatively low population coverage. Therefore, the generation of hypoimmunogenic iPSCs seems to be a more universal approach.

Hypoimmunogenic iPSCs’ engineering is a multi-step process involving the knockout or knockin of several genes [6,7]. The clustered regularly interspaced short palindromic repeats—CRISPR-associated proteins (CRISPR/Cas)—systems are best suited for such a complex task. Currently, the most widely used CRISPR/Cas system for genome editing is the CRISPR/Cas9 *Streptococcus pyogenes* type II-A system. This system consists of a large (1368 aa) multifunctional and multidomain Cas9 nuclease and a small non-coding RNA part, which can be represented either by two molecules, guide RNA (gRNA) and tracer RNA (tracrRNA), or by a single molecule, single guide RNA (sgRNA). Cas9 and gRNA:tracrRNA (or sgRNA) form a complex that searches for genomic targets using a short sequence, the spacer, which is part of the gRNA. For proper target recognition, the system requires a very short three-nucleotide sequence adjacent to the genomic target, the protospacer adjacent motif (PAM). The recognition of the PAM and successful annealing of the gRNA spacer to the target DNA leads to the activation of Cas9 nuclease activity and the formation of a DNA double-strand break (DSB) in the target DNA [8]. In turn, DSBs activate multiple cellular DNA repair pathways to repair DSBs. The most active pathway, non-homologous end joining, results in small deletions and insertions at the DSB site, resulting in gene silencing if the DSB is in the coding region. When donor DNA is provided by the CRISPR/Cas9 system, it could be inserted precisely at the DSB site using the homology-dependent DNA repair pathway [9]. Thus, the coordinated action of the CRISPR/Cas9 system and cellular DNA repair pathways results in the knockout, precise repair of the mutation, or knockin of the target gene. The success of the CRISPR/Cas9 system lies in the relative ease of targeting, requiring the construction of only a short spacer, and the ability to use multiple spacers simultaneously and thereby target multiple regions of the genome simultaneously. However, this system has a number of drawbacks that reduce the safety and efficiency of genome editing.

This review aims to critically discuss novel strategies for the treatment of T1D by cellular products created with the use of CRISPR/Cas9 genome editing technology.

## 2. Etiology, Epidemiology, Diagnosis, and Treatment of Type 1 Diabetes

### 2.1. Etiology and Epidemiology of Type 1 Diabetes

T1D is an autoimmune disease caused by the T-cell-mediated destruction of insulin-producing pancreatic β-cells [1]. There are two subtypes of T1D [9]: 1A—autoimmune, including latent autoimmune diabetes of adults (LADA); and 1B—idiopathic. According to classical theory, the damaged β-cells expose self-antigens to antigen-presenting cells (APC) and thus initiate pathogenic processes through crosstalk with immune cells. But the exact mechanism of the autoimmune response in T1D is still unknown. This process is promoted by an incompletely understood interaction of various factors, such as age, genetic predisposition, and environmental triggers [10]. However, none of these factors is sufficient to develop early screening programs to identify the presymptomatic normoglycemic phase (prediabetes) [11] and to prevent disease progression to the symptomatic one. The typical manifestation of T1D with symptoms of hyperglycemia (polyuria, polydipsia, enuresis, weight loss, and blurred vision), sometimes even with diabetic ketoacidosis occurs with of 90% β-cells destruction [12]. As a result, the diagnosis is not confirmed until the late stages of absolute insulin deficiency when treatment options are limited to hormone replacement therapy and, in some cases, β-cell transplantation.

According to the systematic review and meta-analysis by Mobasseri M. et al., [13] the incidence of T1D in continental subgroups Asia, Africa, Europe, and the Americas was 15, 8, 15, and 20 per 100,000, respectively. The global prevalence of the disease in these regions was also 9.5% (95% CI: 0.07 to 0.12). The modeling study of Gregory et al. [2] showed that 8,42 million people were living with T1D in 2021 and this number is predicted to increase to 13.5–17.4 million people by 2040. The same trend was demonstrated in the global simulation-based analysis for children and adolescents aged 0–19 [14], where the total incidence of childhood T1D cases is expected to reach 476 700 (95% UI 449 500-504 300) in 2050.

The average peak incidence worldwide ranges from 10 to 14 years of age [15]; however, in high-risk populations it shifts towards the 5–9 years age group. [16]. The increasing prevalence of T1D could be explained by several factors, including improvements in the quality and accessibility of medical care, the obtaining of new data from the Middle East and North Africa regions [15], and a decrease in diabetes-related mortality [17]. Nevertheless, the life expectancy of people with T1D remains 24 years less than that of people without the disease, and this is directly correlated with economic status of countries. For example, the remaining life expectancy of a 10-year-old diagnosed with T1D in 2021 ranged from a mean of 13 years in low-income countries to 65 years in high-income countries [2]. It should be noted that the present analysis of global T1D prevalence and incidence has some limitations, the most important of which are different approaches to collecting epidemiological data in different countries.

### 2.2. Genetic Predisposition and Environment Triggers

Genome-Wide Association Studies (GWASs) have identified more than 60 genes that can be involved in the development of T1D, especially those that affect the human leukocyte antigen (HLA) class II alleles (chromosome 6p21.3) [18]. Genetic variation in the HLA genes influences the specific peptides that can be recognized to initiate an immune response. HLA class I proteins (HLA-A, HLA-B, and HLA-C) present endogenous antigens to CD8^+^ (cytotoxic) T-cells, while HLA class II proteins (DP, DR, and DQ) present antigens to CD4^+^ (helper) T-cells [19]. There are more than 35,000 different possible HLA allelic variants [20], the combinations of which, especially at the HLA DR and HLA DQ class II loci, strongly influence the risk of T1D [21]. For example, the combination of HLA-DRB1*04 with DQA1*03:01-DQB1*03:02 (known as DR4-DQ8) increases the risk of disease, while HLA DRB1*04 combined with DQA1*03-DQB1*03:01 does not. The highest risk of T1D is associated with DR4-DQ8 and DR3-DQ2 haplotypes (DRB1*03:01-DQA1*05:01-DQB1*02:01) [22]. Several non-HLA regions appear to be associated with T1D; however, their contribution to disease risk is much smaller. These regions include genes encoding preproinsulin (INS), protein tyrosine phosphatase (PTPN22), IL-2 receptor subunit alpha (IL2RA), protein tyrosine phosphatase receptor kappa (PTPRK), and thymocyte expressed molecule involved in the selection (THEMIS) [22].

Recently, genetic scores have been proposed to assess the combined effects of different genes on the risk of developing T1D. Oram et al. validated the Type 1 Diabetes Genetic Risk Score (T1D GRS), which allows the identification of young adults with diabetes who will require insulin treatment [23]. This tool includes HLA and non-HLA gene T1D-associated single nucleotide polymorphisms (SNPs) that assist in distinguishing type 1 from type 2 diabetes (T2D), monogenic diabetes, and healthy controls [24]. Another group of Redondo et al. demonstrated the utility of T1D GRS in predicting the development and progression of T1D manifestation in autoantibody-positive relatives [25]. Moreover, this tool identified infants without a family history of T1D who had a greater than 10% risk for pre-symptomatic disease, and a nearly two-fold higher risk than children from the high-risk HLA genotypes group alone [26].

There is evidence that environmental factors can also impact the development of T1D. Among potential triggers, much attention is paid to climatic conditions (including sufficient insolation), vitamin D deficiency, and dairy consumption [3]. Presumably, some viral infections (enteroviruses, herpesviruses, rotaviruses, retroviruses, and picornaviruses) might trigger islet autoimmunity, but the direct mechanism remains unknown [26]. The gut microbiome is crucial for immune regulation and could potentially protect against T1D. The large Teddy study of the gut microbiome did not reveal significant differences in patients with islet autoantibodies or manifest T1D. However, the analysis showed that the microbiomes of these patients contained higher numbers of genes involved in fermentation pathways and the production of short-chain fatty acids (SCFAs). Some SCFAs are involved in the mechanisms of gut epithelial integrity maintenance, anti-inflammatory responses, and the regulation of T-cells’ activity. However, further research is required [27].

### 2.3. T1D Diagnosis

The diagnosis of diabetes mellitus is based on World Health Organization laboratory criteria and clinical patterns (see Table 1). However, the cut-offs presented do not include gestational diabetes and do not allow T1D to be separated from T2D. The rate of β-cell destruction in T1D is highly variable. Infants, children, and adolescents often present with diabetic ketoacidosis, while young adults may retain sufficient β-cell function to prevent severe hyperglycemia. A low or undetectable plasma C-peptide level could be used as a marker of endogenous β-cell destruction in the later stages of the disease [28]. However, the lack of standardization between different C-peptide assay methods may make interpretation difficult in cases of slower disease progression, such as latent autoimmune diabetes (LADA) [29].

Autoimmune markers include autoantibodies to insulin (IAA), glutamic acid decarboxylase (GADA), tyrosine phosphatase-like protein IA-2 (IA-2A), and zinc transporter 8 (ZnT8A). GADA can also be detected in non-diabetic patients with autoimmune diseases other than T1D, and does not necessarily reflect insulitis. Conversely, IA-2A and ZnT8A serve as surrogate markers of pancreatic β-cell destruction [30]. The majority of children at risk for T1D who had multiple islet autoantibodies seroconversion progressed to symptomatic disease over the next 15 years [31]. However, the role of autoantibodies in β-cell damage remains uncertain [32]. Patients with idiopathic T1D have no autoantibodies, and some patients with LADA or T2D may have some autoantibodies present, making these tests less specific [33]. Thus, the measurement of autoimmune markers is only recommended in case of the clinical manifestation of T1D, for research purposes, and as a screening for relatives of a proband with T1D [9].

In summary, routine screening programs are not feasible due to several reasons. There is no correlation between the type of autoimmune marker and disease severity. The onset of the symptomatic phase is variable and independent of the time of seroconversion. As no reference values have been established, the clinical significance of further treatment tactics remains uncertain. Overall, the cost-effectiveness of such programs is also controversial [34].

### 2.4. T1D Treatment Options and Complications

To date, there are no drugs that can prevent the development of T1D. There is only one FDA-approved drug that can delay the onset of symptomatic T1D [35]. Teplizumab is a humanized anti-CD3 monoclonal antibody that reduces T-cell autoreactivity [36]. Widespread use of the drug is limited by its high cost (up to USD 200,000 per course) [37] and by immune-related adverse events such as cytokine release syndrome, serious infections, and lymphopenia [38,39].

Daily insulin injections are the standard of care for patients with T1D. This treatment is life-saving, but it is expensive, can lead to acute and chronic complications, and still results in an overall decreased life expectancy [3]. Often, patients with T1D require lower doses of insulin shortly after starting insulin therapy. This phenomenon is commonly referred to as the honeymoon phase [40]. During this phase, the regulation of carbohydrate metabolism depends on the remaining insulin secretion by beta cells, which typically continues for about three months. However, after this period, insulin intensification is required [41]. The traditional basal–bolus regimen could be supported by multiple daily injections or continuous subcutaneous insulin infusion via pump [42]. The regular self-monitoring of glucose is provided by multiple fingerstick blood measurements or by subcutaneously placed continuous glucose monitoring (CGM) [43]. Hybrid closed-loop technology (HCL)—also known as the artificial pancreas—is a relatively new treatment option. It is a system consisting of a CGM, special software to calculate the amount of insulin, and a pump to deliver insulin [44]. Future directions include fully closed-loop and dual-hormone systems (glucagon and insulin or pramlintide and insulin dual-hormone systems). For example, glucagon and insulin dual-hormone systems have been created to reduce both hyperglycemia and hypoglycemia compared to conventional insulin pump therapy. A major drawback is the unstable liquid formulation of glucagon, which requires daily replacement, and the need for two pump systems [45].

Unfortunately, insulin therapy does not mimic physiological processes 100%. In response to a blood glucose increase, β-cells secrete insulin in a biphasic manner with a transient first-phase peak of 5–10 min followed by a more sustained second phase. An important part of this normal insulin secretion is accurate glucose sensing; as blood glucose rises, insulin secretion needs to increase in a dose-dependent manner. Insulin secretion should also become pulsatile in the appropriate glucose range, with little or no pulsation in the fasting state and a larger pulsation after meals. Pulsatile insulin secretion is a hallmark of healthy β-cells, and is caused by fluctuations in their metabolic and electrical activity. Pulsatility suppresses hepatic glucose production and maintains high hepatic glucose receptor sensitivity more effectively than continuous insulin delivery, highlighting the importance of this pulsating regime [46,47].

The regulation of glucose metabolism is provided via multiple-loop control consisting of different signaling mechanisms, crosstalk between all islet endocrine cells, and the modulating effect of incretin hormones produced by the gastrointestinal tract [48]. Exogenous insulins do not act in a biphasic manner, which means the absence of early and late phases of insulin secretion, as well as the interruption of crosstalk with other incretin hormones [49]. The subcutaneous tissue is less well perfused than the β-cells, so insulin injected subcutaneously is absorbed longer. This results in a delayed rise in insulin concentration that does not correlate with increasing levels of glycemia. In other words, the peak of hyperinsulinemia is always out of sync with the peak of hyperglycemia. It is worth noting that the inhaled type of insulin is not widely used among patients with T1D due to its lower effectiveness, high cost, and contraindications in the case of lung disease, smoking, etc. [42].

Daily insulin injections have their own complications. Balancing between intensifying insulin therapy to improve glycemic control and increasing the risk of hypoglycemia is a cornerstone for T1D patients. Clinically important hypoglycemia is below 3.0 mmol/L (54 mg/dL) and is characterized by sweating, weakness, dizziness, and fainting [50]. In severe stages, it alters mental status and is potentially fatal. The long-term treatment of T1D increases the risk of hypoglycemia unawareness [51] due to inadequate sympathoadrenal response to hypoglycemia. Furthermore, improper injection techniques can lead to lipohypertrophy, local allergic reactions that interfere with insulin absorption [52]. In rare cases, the development of antibodies against exogenous insulin can significantly complicate the treatment of diabetes [53].

In general, T1D complications are divided into acute and long-term. Ketoacidosis, severe dehydration, and disorders of consciousness prevail among the acute ones. Long-term complications include microvascular complications and macrovascular disease, and represent a significant economic and treatment burden due to increased patient disability and comorbidity. Microvascular complications include diabetic retinopathy, diabetic nephropathy, and diabetic neuropathy. Macrovascular complications affect the larger blood vessels and include coronary artery disease, cerebrovascular disease, and peripheral vascular disturbances. According to a meta-analysis by Ansari-Moghaddam A. et al., the duration of diabetes has a strong impact on treatment costs (*p* = 0.001) [54] and is also associated with a major cause of diabetes-related complications—poor glycemic control [55]. In line with Gregory P. Forlenza et al. [56], the use of HCL technology significantly improves glucose time within the target range in comparison to sensor-augmented pump therapy (71.0–6.6% vs. 52.8–13.5%; *p* = 0.001). Moreover, continuous insulin delivery via pump has advantages in the management of Mauriac syndrome [57]. However, despite other modern advances in disease control, long-term outcomes still depend on self-management and regular follow-up, as diabetes devices do not operate autonomously.

The key approach to preventing or delaying the progression of complications is to minimize modifiable risk factors. For instance, reducing HbA1c variability [58] and using drugs with nephroprotective effects [59] could delay the onset and progression of diabetic nephropathy. Unfortunately, there is no effective therapy to repair the nerve damage and induce the regression of diabetic neuropathy. Diabetic foot syndrome remains a major cause of lower limb amputation, resulting in a significant reduction in life expectancy (on average only 2 years from the amputation) [60]. The traditional treatment of diabetic foot is mainly medical treatment and the surgical reconstruction of blood flow. Recently, efforts have been made to develop innovative and effective therapies to repair chronic wounds, including the topical application of growth factors or cell-based therapies [61]. In addition, the treatment options of painful diabetic neuropathy have been significantly improved [62]. A new prospective direction has also been developed in the therapy of diabetic retinopathy. There is a large number of new intravitreal anti-vascular endothelial growth factor injections [63], which are successfully used along with classical treatment methods (laser photocoagulation and vitreoretinal surgery).

Despite advanced technology, the prevalence and incidence of T1D complications remain significantly high, leading to a decline in patients’ quality of life. Only the development of a new paradigm in treatment that leads to a complete cure will represent a new milestone in medicine.

### 2.5. T1D Cell Therapy

Researchers have long been working on cell therapy for the treatment of patients with T1D, including the transplantation of Islets of Langerhans [64]. All known methods are aimed at introducing glucose-sensitive cells capable of producing insulin into the patient’s body. Recently, the first FDA-approved cell therapy for T1D was reported. Lantidra (donislecel) (CellTrans Inc., Chicago, IL, USA) is a therapy with pancreatic islet cells derived from the pancreatic cells of deceased donors. Participants received 1 to 3 infusions of allogeneic islet β-cells. Of the 30 patients, 21 had not received insulin for at least one year, 11 had not needed insulin for one to five years, and 10 had not received insulin for more than five years. Five patients failed to achieve a single day of insulin independence (ClinicalTrial NCT03791567). A significant limitation of this method of therapy is the discrepancy between the number of donors and recipients, which makes it impossible to implement it widely in clinical practice.

To overcome this limitation, researchers have turned to stem cells, which, depending on the source, may allow more patients to be treated. Several other cell therapies are currently being developed for the treatment of T1D (Table 2). Data from the ongoing Phase I/II study of VX-880 (Vertex Pharm., Boston, MA, USA) show that two patients achieved insulin independence with HbA1c values of 5.3% and 6.0% after at least one year of treatment with the drug (ClinicalTrial NCT04786262). VX-880 is an allogeneic therapy with islet cells derived from stem cells that are fully differentiated and produce insulin. VX-880 is administered through infusion into the hepatic portal vein. Immunosuppressive regimens are used to prevent cell rejection. Both Lantidra and VX-880 involve this kind of treatment. The drug VX-264 (Vertex Pharm., Boston, MA, USA) takes a different approach, using the same pancreatic islet stem cells as in VX-880, encapsulating the latter in a surgically implantable canal–arterial protective device to protect them from the recipient’s immune system (ClinicalTrial NCT05791201). VX-264 has recently received FDA approval as an Investigational New Drug Application. Although this approach avoids the need for immunosuppression, it has its challenges. Encapsulated devices can fibrosize, and physical separation from the bloodstream negatively affects the survival of stem cell-derived islets (SC-islets) and may limit their ability to respond to blood glucose levels. This was the problem encountered by Viacyte (South San Francisco, CA, USA) in the first clinical trial of the VC-01 device encapsulating pancreatic endoderm that has completed its differentiation into functional islet cells after subcutaneous transplantation (ClinicalTrial NCT02239354). In a subsequent trial, the device was fitted with portals to allow vascular ingrowth (ClinicalTrial NCT03163511). Direct vascularization increases cell survival by improving oxygenation and metabolic metabolism, but opens access to the immune system, resulting in the need for pharmacological immunosuppression to prevent rejection [65]. Another disadvantage of therapy based on the maturation of pancreatic endoderm cells in the body after transplantation is the lack of control over the direction of differentiation, which in turn is highly variable. The immunostaining of the graft revealed an abundance of glucagon-positive cells, while insulin-expressing cells were a minority [66].

Sernova Corp. (London, ON, Canada) has taken a different approach by introducing a new implantable device called Cell Pouch. The device is a scaffold made of non-degradable polymers, formed into small cylindrical chambers, which, when implanted into the abdominal muscle, grow vascularized tissue around the perimeter of removable plugs in as little as two weeks. After tissue engraftment, the plugs are removed, leaving fully formed tissue chambers with central cavities for the transplantation of therapeutic cells such as insulin-producing islets. The Cell Pouch forms a natural environment rich in microvessels that allows transplanted islets to engraft. According to data presented from a Phase I/II trial of the Cell Pouch system in patients with T1D, the first 5 patients who underwent islet transplantation achieved insulin independence for 6 to 38 months (ClinicalTrial NCT03513939).

An undoubted achievement of T1D cell therapy is the emergence of combination approaches using gene-edited allogeneic stem cells. A phase I/II trial of a combinatorial drug, VCTX211 (CRISPR Therapeutics AG (South Boston, MA, USA) in collaboration with Viacyte), consisting of two components, allogeneic pancreatic endoderm cells genetically modified with CRISPR/Cas9, and a removable perforated device designed to deliver and preserve these cells, is currently underway (ClinicalTrial NCT05565248). VCTX211 has two gene knockouts (*B2M*, *TXNIP*) and four insertions (*PD-L1*, *HLA-E*, *TNFAIP3*, and *MANF*) to improve functionality. Unlike the previous version of the drug, called VCTX210A, whose clinical trials have already been completed (ClinicalTrial NCT05210530), the current version of VCTX211, in addition to edits to reduce T- and NK-cell mediated immune rejection and protection against oxidative stress in the endoplasmic reticulum, was supplemented with the A20 (*TNFAIP3*) gene insert, which induces graft engraftment and protection against cytokine-induced apoptosis, and with the MANF gene insert, which enhances β-cell proliferation and protection against inflammatory stress.

#### 2.5.1. Sources for β-Cell Generation

Research on the production of insulin-producing cells in vitro has been underway since the opportunities offered by human pluripotent stem cells (PSCs) for regenerative medicine became clear. Sources of islet endocrine clusters for in vitro production of insulin-producing cells include embryonic stem cells (ESC), iPSCs, adult stem cells, and differentiated cells from mature tissues that can be transdifferentiated into insulin-producing cells.

Intensive research has been conducted to develop protocols for generating insulin-producing cells from stem cells, from the first seminal work describing the generation of definitive endoderm [67] to the creation of stem cell-derived β-cells that secreted insulin in response to successive glucose challenges [68] and the generation of monohormonal, insulin-expressing β-cells [69,70,71,72]. Present strategies for SC-islets are predominantly based on approaches that imitate normal pancreas development [73]. A common principle for the various multi-stage protocols is that iPSCs cultured in 3D are differentiated by targeting key embryonic signaling pathways such as Nodal, WNT, RA, FGF, BMP, Notch, and Hippo (Figure 1). On average, this takes five to seven stages that last 20-30 days [74], but functional maturation associated with transcriptional maturation can continue for months after cell transplantation [75]. Recently, more than 600 genes have been identified whose expression increases 6 months after cell transplantation in vivo [76].

The latest generation protocols describe the derivation of islet clusters from stem cells, in which other endocrine cell types such as α- and δ-cells are present along with the target β-cells. The role of α- and δ-cells in terms of β-cell function in the context of islet clusters, as well as their optimal ratio, currently remains unknown. Intestinal endocrine cells, termed enterochromaffin cells, are one of the most common off-target cell types in islet clusters. Other off-target cells can be hepatic, mesenchymal, and pancreatic exocrine cells [69]. The presence of such off-target cell populations may negatively affect the functional activity of β-cells [69]. In addition, although modern generation islet clusters are capable of secreting significant amounts of insulin in response to glucose stimulation in vitro, there are still fundamental transcriptional and epigenetic differences between artificially derived islet clusters and primary human islets. Since these differences primarily affect the activity of metabolic pathways and the functionality of islet clusters, studies are underway to better characterize cells derived from differentiation protocols. Multiomic studies show that β-cells derived by in vitro differentiation have the least similarity to their primary counterparts compared to other endocrine cell types [76,77]. Moreover, the chromatin of primary human islet cell types is more restricted than in islet-like clusters obtained in vitro, where the *INS* gene remained open not only in β-cells but also in α-cells and δ-cells [76]. However, the long-term culturing of SC-islets both in vitro and in vivo results in a transition of chromatin to a more closed state. By comparing scRNA-seq and snATAC-seq data, it was found that transplantation directs cells to their correct identity, and SC-islets become more similar to primary human islets compared to those formed in vitro [77]. Thus, the detailed characterization of SC-islets using single-cell sequencing technologies sheds light on the unknown mechanisms controlling islet development and currently limiting islet maturation. The further optimization of differentiation protocols is needed to obtain fully mature islets with a minimum number of potentially unsafe off-target cell types in vitro and thereby increase the safety of transplantation of in vitro differentiated islets to patients.

Transdifferentiation into β-cells can occur from a variety of sources including hepatocytes, gastrointestinal cells, exocrine pancreatic cells, and other endocrine cell types. Since these cells share common precursors with β-cells and therefore similar developmental pathways and epigenetic profiles, transdifferentiation from these cell types is most appropriate [78]. There are two different approaches to the transdifferentiation of cells: through genetic modifications in the cell or by exposing cells to certain molecules to activate certain signaling pathways. One of the most common ways to obtain insulin-producing cells is the exogenous overexpression of certain transcription factors.

In 2008, Zhou et al. first identified three key transcription factors (MafA, Pdx1, and Ngn3) that induce the reprogramming of fully differentiated exocrine cells into β-cells, demonstrating the possibility of targeted reprogramming of adult cells without reverting to a pluripotent state [79]. Pdx1, also known as pancreatic and duodenal homeobox transcription factor 1, is required for β-cell maturation and the maintenance of their metabolism and proliferation. MafA binds to an enhancer that regulates insulin gene expression in pancreatic cells. Neurogenin-3 (Ngn3) is a transcription factor common to endocrine progenitors, so all endocrine cells originate from an Ngn3-positive progenitor.

Since α-cells are the second most abundant type of pancreatic endocrine cell that can be transformed into insulin-producing cells, the greatest amount of research has focused on studying transdifferentiation from α-cells. In murine models of diabetes, it has been shown that α-cells increase their size and proliferation rate during the development of the disease, while the level of apoptosis is significantly reduced [80]. Moreover, the number of glucagon/Pdx1-positive cells increases in these mice during diabetes, indicating a possible transition of α-cells to β-cells in vivo. Given these facts and the common origin of α- and β-cells, the transdifferentiation of α-cells may seem to be a promising approach to the treatment of diabetes.

Additionally, there have been reports of transdifferentiation possibilities in vivo, in addition to attempts made to create insulin-producing cells in vitro. Recently, it was demonstrated that α-cells can be transdifferentiated into β-cells in mice in vivo by delivering Pdx1 and MafA with a promoter specific for human α-cells using adeno-associated virus (AAV). Transduction was performed on mice with chemically induced and autoimmune diabetes, and resulted in decreased hyperglycemia in both cases. An increase in proliferation rate was also proven to enhance the effect of viral transduction and help prevent further disease progression [81]. Thus, the resulting insulin-producing cells were not really β-cells but β-like cells, because they expressed specific markers of the two cell types.

Except the attempts of creating insulin-producing cells in vitro, opportunities of in vivo transdifferentiation were also reported. Recently, it has been demonstrated that α-cells can be transdifferentiated into β-cells in mice in vivo via delivering Pdx1 and MafA with human α-cell specific promoter using AAV. The transduction was held in mice with chemically induced and autoimmune diabetes and resulted in a decreased hyperglycemia in both cases. An increased proliferation rate was also proven to enhance the effect of viral transduction and contributed to the prevention of the further development of the disease [82]. Thus, the in vivo transdifferentiation approach showed higher efficiency, which may be a consequence of the presence of a microenvironment and certain endogenous signals.

Similar strategies of exogenous overexpression were also applied to other cell types. For example, the simultaneous overexpression of MafA, Pdx1, and Ngn3 allowed the transformation of Sox9-positive mouse liver cells into insulin-secreting cells [83]. However, the transdifferentiation of hepatocytes has limitations due to their low proliferative capacity [84]. To increase the efficiency of transdifferentiation, the researchers used human liver stem cells, which have shown the ability to form islet-like structures. Although the resulting islet-like structures were immature, they displayed an insulin response to glucose and controlled hyperglycemia in murine models of diabetes [84].

Along with transdifferentiation protocols based on epigenetic modifications, there are approaches to induce the transition into insulin-producing cells by exposing them to specific inhibitors and growth factors. Non-endocrine pancreatic cells can differentiate into β-like cells through the simultaneous inhibition of TGFb and activation of the BMP pathway through exposure to BMP7. However, the resulting cells lacked the necessary features and were unable to maintain glucose homeostasis after in vivo transplantation [85]. Thus, the method of exposing cells to specific molecules was not quite effective.

Currently, there are no effective transdifferentiation protocols for obtaining mature β-cells, responding to glucose by secreting the required amount of insulin. However, inducing the overexpression of specific transcription factors has been shown to be more effective than exposing cells to low molecular weight inhibitors and inducers. In addition to incomplete β-cell differentiation, another challenge for T1D therapy may be allogeneic cell-derived graft rejection. In this respect, patient-specific cell products derived from autologous PSCs are an ideal option for transplantation. However, allografts are preferable to autografts, both in terms of cost and the speed of obtaining them. After all, the use of cell products in medicine involves the collection of thorough preclinical data for each individual cell line, which is associated with the significant expenditure of material resources and time. Thus, it seems reasonable to use allogeneic cell products, but then the problem of the rejection of transplanted cells becomes relevant. Currently, there are two approaches to its solution: the creation of biobanks of iPSC lines allowing the needs of the majority of patients to be covered, and the creation of immunoprivileged iPSC lines.

#### 2.5.2. HLA Haplotype Banks

HLA-haplotype banks are now widely used throughout the world. Bone marrow and cord blood samples with different HLA-haplotypes are used in hematopoietic stem cell transplantation for patients suffering from leukemia and other blood diseases to minimize the impact of immune rejection. Therefore, one approach to reduce the cost of iPSC-based therapy is to use allogeneic donors homozygous for common HLA types. This approach does not require genome editing and allows the generation of iPSC lines and the creation of a clinical-grade iPSC bank covering a large percentage of the population.

In the case of the Japanese population, 30 iPSC lines homozygous for common HLA types selected from 15,000 donors were assumed to provide a three-locus (loci affecting graft rejection—HLA-A, HLA-B, and HLA-DR) match for 82.2% of recipients, whereas 50 iPSC lines selected from 24,000 donors provided coverage for 90.7% of the same population [4]. Okita et al. reported that the screening of 160,000 donors resulted in the selection of 140 HLA-homozygous donors with iPSCs, providing a match for 90% of the Japanese population [5]. Analytical data based on prospective HLA typing of large numbers of individuals in other ethnic populations are available. It has been estimated that 26,000 donors for European Americans and 110,000 donors for African Americans would need to be screened to achieve a match of 50% and 22% of the recipient population, respectively [86].

In contrast to previous approaches based on randomly finding HLA types, Taylor et al. proposed an alternative approach based on the concept that iPSCs derived from adult donors with the desired homozygous HLA type could be identified from existing registries of voluntary stem cell donors. By calculating all theoretically possible homozygous HLA-A, -B, and -DR combinations, 405 combinations were found that together could provide an HLA match for all potential UK recipients. It was found that the most useful homozygous HLA-types present among the more than 17 million HLA-typed potential stem cell donors registered on the Bone Marrow Donors Worldwide website could provide a complete HLA match for 93.16% of the UK population [87]. HLA–antigen matching rates are similar in countries with similar demographic majorities. The top four homozygous haplotypes found in the Australian population are identical to the top four haplotypes found in the United Kingdom [88]. Similar modeling in Korea also revealed common homozygous HLA haplotypes with most Asian countries including Japan, China, Hong Kong, Taiwan, Vietnam, the Philippines, etc. [89].

At the same time, comparative HLA analysis shows that iPSC lines relevant in one country have limited use in ethnically diverse populations in other countries, although in some cases there is significant overlap in certain HLA haplotypes. The HLA-haplotypes found by Okita et al. (HLA-A*24:02; HLA-B*52:01; HLA-DRB1*15:02 and HLA-A*11:01; HLA-B*15:01; and HLA-DRB1*04:06) are present in 8.5% and 1.3% of the Japanese population, respectively, and iPSCs derived from these two donors match approximately 20% of Japanese individuals [5]. However, theoretically, these HLA haplotypes may not be of much use in, for example, the UK population, as none of the haplotypes mentioned are part of the optimal combination of the most useful homozygous HLA types reported by Taylor et al. [87].

In addition, HLA matching with homozygous donors does not guarantee protection against NK-cell attack. There are two HLA-C groups, namely HLA-C1 and HLA-C2, which bind to different inhibitory receptors, KIR2DL3 and KIR2DL1, respectively, on the surface of NK-cells [90]. When using donor HLA homozygous iPSC lines, either C1/C1 or C2/C2, the phenomenon of the lack of self-recognition due to lack of inhibitory signaling will be observed for recipients with C1/C2. In the Japanese population, the frequency of C1 and C2 allotypes is 92.7:7.3. Therefore, the above homozygous iPSC lines would be applicable and mismatches would be rare. However, in other populations where the frequency of C2 is higher, this issue will be more significant. For example, in the Polish population, the ratio of C1 to C2 is approximately 6:4 [91], as well as in Russia [92].

The heterogeneity of ethnic populations indicates that there is no single universal set of the most useful homozygous HLA-types to maximize the coverage of the entire world. Each country selects optimal sets based on the prevalence of certain HLA haplotypes in its population. The World Marrow Donor Association currently estimates that 804,522 cord blood units are stored in cord blood banks worldwide [93]. Collaboration with established biobanks allows the efficient identification of HLA-homozygous donors to prepare a stock of iPSC lines with the best immune compatibility in a given population [88,94,95,96]. Therefore, it should be understood that the establishment of a global network of iPSC banks is necessary to lay a solid foundation for future worldwide clinical cell therapy. At the same time, even with the selection of HLA-adequate donor material, immunosuppressive therapy after transplantation cannot always be completely avoided. However, this approach allows the dose and duration of immunosuppression to be reduced, which in itself is a significant advantage for patients.

### 2.6. Engineering of Immunoprivileged iPSC Lines

Human clinical trials show that progenitor cells (pancreatic endoderm cells), as well as fully differentiated insulin-producing cells derived from iPSCs, can be safely implanted in patients with T1D. There are currently no proven stem cell therapies for diabetes, but clinical trials are underway to confirm the presence of functional activity of pancreatic progenitor cells or iPSC-derived β-cells after transplantation. The success of these trials will point the way to solving the problem of obtaining the necessary number of functional islets for transplantation.

Genome editing technology offers an alternative strategy for stem cell-mediated diabetes therapy using hypoimmunogenic human iPSCs. HLA mismatch is a major cause of T-cell or antibody-mediated graft rejection [97,98,99]. It has previously been shown that the inactivation of HLA-A genes using artificial zinc finger nucleases allows modified hematopoietic stem cells (HSCs) to retain the ability to engraft and restore hematopoiesis in a mouse model [100]. The results of these experiments suggest the possibility of transplanting other engineered clinically relevant cell types, such as iPSCs derived from a single donor, to multiple recipients [101]. Unlike HLA-haplotype banks, the main advantage of this approach is the small number of lines required to cover the entire global human population. It is estimated that the production of a clinical-grade cGMP-compliant human iPSC line would cost between USD 200,000 [102] and USD 800,000 [103]. It should be taken into account that each iPSC line is unique in morphology, epigenetic status, growth curve, gene expression, and, most importantly, differentiation potential. This heterogeneity is one of the significant limitations of cell therapy that needs to be overcome, hence the current preference for using a single line rather than multiple lines.

The creation of immunoprivileged iPSCs requires an understanding of the molecular interactions between cytotoxic immune cells and foreign cells in order to eliminate those molecules and add those that allow transplanted cells to avoid death from cytotoxic immune cells.

HLA class I molecules are expressed in most cell types and consist of a main chain unique to each HLA protein and a β2-microglobulin chain common to all class I complexes. They are recognized by a complex consisting of a T-cell receptor (TCR) and CD8, which is present on CD8+ T-cells, also known as T-killers or cytotoxic T-cells. They destroy a cell presenting foreign peptides as they are potentially infected. HLA class II are expressed on the surface of APCs such as dendritic cells, macrophages, and B-cells. They consist of two identical α and β chains, and are recognized by a complex consisting of TCR and CD4 located on the surface of CD4+ T-cells, or T helper cells, resulting in the activation of the immune response [104]. The absence of any HLA on the cell surface may lead to its destruction by natural killer (NK) cells on a “lack of one’s own” basis, since there is a high probability that such a cell is infected with a virus, which thus tries to hide its peptides from exposure, or is carcinogenic [105].

The immune rejection of transplanted cell products, tissues, and organs due to donor-recipient HLA mismatch is one of the major problems of transplantation. HLA class II is not usually considered in the context of this situation because it is present in only a few very specific cell types, in contrast to the ubiquitously expressed HLA class I. The high polymorphism of many genes encoding proteins in this group was probably evolutionarily necessary to protect against transmissible tumors, such as those found in Tasmanian devils, dogs, and some bivalves [106]. Indeed, classical *HLAs*, which include the *HLA-A*, *HLA-B*, and *HLA-C* genes, are highly polymorphic, while non-classical *HLAs*, such as *HLA-E*, *HLA-F*, and *HLA-G*, are oligomorphic [107,108,109,110]. In addition, the gene encoding β2-microglobulin is virtually monomorphic (Figure 2).

Thus, one of the most ideologically simple ways to evade T-cell cytotoxicity is to completely eliminate all HLA class I molecules by knockout of the gene encoding β2-microglobulin (Figure 2). However, this led to the destruction of the allograft by NK-cells, which could not bind to HLA and make sure of its presence and that the cell was healthy. [111,112,113]. Another strategy is to simultaneously knockout the classical highly polymorphic *HLAs* genes while retaining the low polymorphic non-classical *HLA-E*, *HLA-F*, and *HLA-G* genes, resulting in the increased survival of engineered cells (Figure 2) [114]. In addition to the *HLA-E*, *HLA-F*, and *HLA-G* genes, the *HLA-A2* allele can be retained [115]. *HLA-A2* is an extremely common allele in the human population, its frequency among some ethnic groups being about 50%, so the retention of this allele makes such cells histocompatible with a large group of people [108,115]. A similar strategy was used by Ji et al. [116]; they knockouted *HLA-B* and *-C* in hESCs while preserving *HLA-A*11:01*, which is present in 21% of the Chinese population. Another way to protect engineered cells from NK-cells is to insert genes encoding non-classical HLAs into *B2M* knockout cells. Thus, Chen et al. [7] obtained *B2M*^−/−^ iPSCs with *HLA-G* overexpression via lentiviral transduction. Gornalusse et al. [114] made an *HLA-E* knockin at the *B2M* locus of one of the alleles, so that *HLA-E* was fused to *B2M* and the second *B2M* allele was disrupted. Modifications were carried out through nuclease-free editing using recombinant AAVs. It has also been shown that the protection of *B2M* knockout cells from NK-cell attack can be achieved by overexpression of *CD47* [117].

Although HLA-II is expressed mainly on antigen-presenting cells, its expression can be observed on other cell types during inflammation and in autoimmune diseases [118]. In conclusion, it has been shown that β-cells of T1D patients express HLA-II more frequently than healthy individuals [119]. Thus, when creating iPSCs with reduced immunogenicity, most studies suppress HLA-II expression through the knockout of the *CIITA* gene, which is a major activator of *HLA-II* genes [113,115,116,120,121,122,123,124,125].

In addition to *HLA* genes manipulation, there are a number of other genetic modifications that can increase the viability of β-cell allografts. First, the overexpression of PD-L1 molecule, which binds to the PD-1 receptor on the surface of T-cells and transmits inhibitory signals, further reduces cytotoxicity [121,126,127]. Second, β-cells are engineered to secrete cytokines (IL-2 mutein, IL-10 and TGFβ) that create a microenvironment that prevents xenograft rejection [128]. Finally, the oral administration of RNLS inhibitor or CRISPR/Cas9-mediated knockout of *RLNS* gene in β-cells resulted in increased graft viability due to protection from autoimmunity [129].

#### 2.6.1. Safety Concerns and Approaches to Overcome Them

Currently, the expectations for realizing the opportunities offered by ESC and iPSC technologies are higher than ever. However, one of the major limitations that must be overcome for their widespread use is oncogenicity due to the ability of these cells to proliferate in a virtually unlimited way [102,130]. This occurs for a number of reasons. First, undifferentiated iPSCs may form teratomas after transplantation to a recipient, so it is necessary to develop effective methods for the directed differentiation of iPSCs in vitro and/or methods of purification of the final cellular product from the residual amount of undifferentiated cells. Moreover, even if stem cells differentiate into the desired type, this process is not complete, and their chromatin remains largely immature. Second, tumorigenesis may be promoted by residual activity of reprogramming factors in iPSCs. Reprogramming factors are often associated with oncogenicity, especially c-Myc, which is one of the most frequently mutated genes in human cancer cells, and for this reason, the possibility of replacing it with the less oncogenic L-Myc is being considered [131]. Third, when iPSCs are cultured in vitro prior to transplantation, genetic alterations such as chromosomal aberrations, copy number variations, and single-nucleotide mutations may occur, which in turn may increase the risk of cancer, whereas in the case of iPSCs, pre-existing somatic mutations may also be present. Finally, clones with higher tumorigenic potential are more competitive and thus more likely to be selected in cell culture screening. [102,130].

Therefore, there is a need to develop strategies to effectively monitor the safety of a cell therapy product. Some of these can be drawn from the experience with chimeric antigen receptor (CAR) T cells, as they are one of the most advanced cell therapies to date—they are used against both solid tumors and lymphomas and are currently in phase I to III clinical trials [132,133,134].

In the following, we briefly review some of the currently known strategies for controlling CAR-T cells [134,135,136], which may be applicable to β-cells as well. Approaches with the selective killing of target cells are considered to be the most effective. In the case of CAR-T therapy, this avoids immunotoxicity, and in the case of β-cells it will control their potential cancerous transformation and hormone hyperproduction. Approaches with target cell destruction can be divided into three groups (Figure 3).

The first group of strategies is based on the use of monoclonal antibodies and antibody-dependent cell-mediated cytotoxicity (ADCC). For example, a truncated EGFR (tEGFR) may be expressed on the surface of cells. Since such a protein does not contain part of the cytoplasmic domain, it does not affect cellular metabolism, and yet the antibody cetuximab can be used against cells expressing tEGFR [137,138,139]. In the case of the expression of the artificial *RQR8* gene encoding the antigens CD34 (which is used for positive selection of cells containing *RQR8*) and CD20, cell elimination is achieved using the anti-CD20 antibody rituximab [140,141,142,143]. Finally, it has been proposed to introduce a 10-amino acid sequence derived from c-Myc into a recombinant TCR and then use an antibody against c-Myc. A disadvantage of this particular target is the lack of clinical trials to date, as there is no clinically approved anti-C-Myc antibody [144]. The disadvantages of the entire strategy include (1) the fact that cetuximab and rituximab cannot cross the blood–brain barrier [145,146], so they will be ineffective in brain metastases; (2) slightly lower efficacy (82.5% for the cetuximab-tEGFR system) compared to the approaches discussed below [147], at least for CAR-T cells; and (3) the fact that both EGFR and CD20 are sometimes targets of immunotherapy [148,149,150], so if a patient develops a cancer that requires a course of immunotherapy with cetuximab or rituximab, the modified β-cells will also die.

The second group of strategies uses a prodrug that the transgenic enzyme converts into an active compound that causes cell death. The most common such enzyme is herpes simplex virus thymidine kinase (HSV-TK). The currently used form of this enzyme is called HSV-sr39TK and contains five mutations that increase its efficiency [151]. When administered with ganciclovir (GCV) (acyclovir or penciclovir can also be used); HSV-TK kills the cell by a mechanism that is not fully understood. It is only known that HSV-TK phosphorylates GCV or its analog, and cellular kinases further form a triphosphate derivative that is incorporated into DNA during replication, resulting in the suppression of DNA synthesis and cell death [152,153]. Of note, a membrane-bound version of HSV-TK attached to CD34 or NGFR has been developed [154,155,156]. However, the most serious disadvantage of such a system is its high immunogenicity, since HSV-TK is not normally expressed in human cells [157,158]. This can lead to immune rejection of HSV-TK expressing β-cells. In addition, this system targets only actively proliferating cells, and is ineffective against resting/quiescent cells. To overcome these limitations, a mutant human thymidylate kinase (TMPC) was obtained, which phosphorylates 3′-azido-3′-deoxythymidine (AZT), and the resulting product leads to disruption of the inner membrane potential of mitochondria and activation of caspase 3. Thus, the TMPC/AZT combination, unlike HSV-TC/GCV, is non-immunogenic and is able to affect both dividing and quiescent cells, which guarantees the complete elimination of the entire population of target cells [159]. However, the safety and efficacy of this system has not yet been confirmed by clinical trials.

The third and apparently most promising approach is the chemically induced dimerization of modified caspase 9 (named iC9). Instead of a natural dimerization domain, iC9 possesses an FKBPv domain from the human FKBP12 protein with the F36V mutation. FKBPv is capable of binding AP1903, also known as rimiducid, and AP20187, symmetrical organic compounds used as homodimerizing agents. The homodimerization of iC9 through FKBPv domains leads to its activation and triggers apoptosis [160,161,162,163,164]. In another modification of caspase 9, iRC9, one part of caspase 9, contains the wild-type FKBP domain and the other part the FKBP-Rapamycin Binding (FRB) domain from mTOR. Such a system heterodimerizes upon rapamycin addition, which also leads to iRC9 activation and cell death [165]. Since all the protein domains used are present in the human body, such a system is not immunogenic [166,167]. It is also very fast and kills most of the population of modified CAR-T cells within 30 min [160]. Therefore, we believe that this system is the most promising for biocontrol of engineered β-cells. However, whether this or any of the other systems discussed here will be truly safe for humans and capable of destroying abnormal β-cells remains to be seen in further studies and clinical trials.

#### 2.6.2. Challenges in CRISPR/Cas9-Based Generation of Immunoprivileged iPSCs

To date, as described above, successful cell genome editing strategies have been developed, leading to the creation of hypoimmunogenic iPSC cell lines, which in the future may become a universal source of “off-the-shelf” cells for allogeneic cell therapy. Gene knockouts to create hypoimmunogenic iPSCs are most often performed using the CRISPR/Cas9-based genome editing system (Table 3). This technology allows making the necessary changes in the genome of cells quite easily and successfully. However, this technology has a number of limitations.

First of all, this is a high off-target rate, which is especially characteristic of *S. pyogenes* Cas9 [177,178]. Off-targets carry potential risks of dysfunction of important genes and associated signaling pathways and metabolic processes, which, in turn, might lead to decreased cell fitness, aberrant phenotype, or possible tumorigenesis. Since several (more than three) alterations are required to produce hypoimmunogenic iPSCs, such multi-target editing increases the likelihood of a large number of off-targets. The presence of a large number of off-targets, including chromosomal translocations in CRISPR/Cas9 edited cells, was found, for example, in two recent studies [179,180]. Notably, despite the presence of off-targets, edited hematopoietic cells transplanted into rhesus macaques [179], as well as edited T-cells transplanted to tumor patients [180], did not exhibit any aberrant phenotype, including cancer transformation, throughout the entire follow-up time (2 years and 9 months, respectively). The data suggest cautious optimism that not all adverse effects of Cas9 may lead to an aberrant phenotype of edited cells, or that such cells are rapidly eliminated by the recipients’ immune system. Nevertheless, the risks of aberrant phenotypes of edited cells still exist, so control of the number of off-target cells is necessary. The variety of in silico and experimental methods used to detect off-targets is discussed in recent reviews [177,181]. It should be noted that each off-target detection method has specific features that allow it to detect some types of off-targets well, but be insensitive to other types. For example, the common method GUIDE-seq [182] and its derivative GUIDE-tag [183] are good at detecting Cas9-induced DSBs, but these methods are ineffective for detecting chromosomal aberrations (translocations, transversions, and large deletions). Hence, it is necessary to use a combination of several methods to see the full possible spectrum of Cas9-mediated off-targets.

Another strategy to control Cas9-induced off-targets is to use its high-fidelity variants. However, in the case of iPSCs, only a small number of studies utilize Cas9 variants with increased specificity (Table 3), such as Hi-Fi Cas9 [128,176,184,185], also in one of the papers the authors used Cas9n nickase [111], a known high-fidelity Cas9 modification [186]. Lamothe at al., in their work, demonstrated the successful use of novel CRISPR-editing systems based on modified type II and type V nucleases for the genetic engineering of primary immune cells and iPSCs [187]. In addition, several high-fidelity Cas9 variants are created, e.g., eSpCas(1.1) [188], Cas-HF1 [185], Hypa-Cas9 [189], SuperFi-Cas9 [190], TurboCas9 [191], SpartaCas [192], LZ3 Cas9 [193], etc., thus expanding the tools for safe genome editing. However, highly specific variants often have lower activity compared to wild-type Cas9 [194,195]. Combined with the low transfection efficiency of iPSCs [196,197], this results in a very low overall genome editing efficiency. To address this problem, several next-generation Cas9-based editors with enhanced activity and high specificity have been developed, OptiHF-SpCas9 [198], eiSpCas(L1206P), SniperCas(L1206P) [199], rCas9HF [200], Sniper2L, and Sniper2P [201]. Although they show promising results, their enhanced activity needs to be tested on a wider range of genomic targets.

The level of Cas9 off-target activity also depends on the gRNA spacer sequence. Hence, it is clear that the design of the spacer makes it possible to reduce the level of off-target activity. There is a family of programs that allow the estimation of the number of potential off-targets (e.g., Cas-OFFfinder [202], as well as the activity of spacers and selecting the most active and specific ones (e.g., CRISPOR [203], CHOP-CHOP [204]).

Highly specific nucleases often have lower activity compared to wild-type Cas9 [194]. This is also a major problem exacerbated by the low transfection efficiency of iPSCs [196,197]. The genetic modifications of iPSCs are carried out using both viral and non-viral delivery methods. At the same time, viral delivery methods using retro- and lentiviruses are unsafe due to their integration mutagenesis, which can lead to cell malignization [205,206,207,208,209,210,211]. The AAVs commonly used in gene therapy have low packaging capacity, which may also limit their use, and there is still a risk of integration into the genome [212,213,214]. Although in many cases this integration does not lead to oncogenesis [213], several cases of this have been described for AAV2 [215,216,217]. A high percentage (47%) of incorporation of the AAV fragment into the on-target site was also found when AAV was used to deliver the CRISPR/Cas9 system. Although no off-target incorporation of AAV has been observed [218], the safety of using AAV for CRISPR/Cas9 delivery in clinical practice remains controversial.

Among non-viral delivery methods, nucleofection and lipofection are the most popular, with nucleofection being the most effective [196,219]. Non-viral delivery methods are simpler and cheaper, but highly toxic to iPSCs and many primary cell lines [220,221]. When creating cells with reduced immunogenicity, the CRISPR/Cas9 system is most often introduced into cells in the form of plasmid DNA (pDNA) or an RNP complex of Cas9 protein and gRNA (Table 3). At the same time, the transfection of cells with plasmid DNA (pDNA) is always associated with the risk of spontaneous integration into the cell genome [222,223]. In addition, standard laboratory methods of pDNA isolation do not ensure the complete purification of preparations from impurities of genomic DNA, RNA, and endotoxin [224], which may contribute to toxicity. Jing et al. showed that the toxicity observed during pDNA nucleofection of T-cells is due to activation of the cytosolic cGAS-STING (cyclic guanosine monophosphate-adenosine monophosphate (cGAMP) synthase-stimulator of interferon genes) pathway, which leads to the activation of the interferon response, resulting in cell death [220]. Another side effect associated with the transfection of cells with pDNA carrying the CRISPR/Cas9 genome editing system is the integration of pDNA or bacterial genomic DNA fragments into edited loci [225], which can lead to the unpredictable disruption of the architecture of the edited region of the genome. A safer but labor-intensive approach in this context is the introduction of RNA or RNP complexes. The use of RNA instead of pDNA or RNP during the transfection of iPSCs and primary cells results in reduced toxicity [226,227,228] and increased electroporation efficiency [227,229,230]. Moreover, CRISPR/Cas9 delivery by RNP nucleofection compared to pDNA nucleofection has been shown to result in a significant reduction in the number of off-targets, which Kim et al. believe is due to the decreased exposure time of the CRISPR/Cas9 system in the cell [229].

In addition to the introduction of knockouts of various genes, the genetic engineering of hypoimmunogenic cells involves the introduction of various knockins. The efficient insertion and expression of transgenes is achieved using different approaches: lentiviral particle [117,231], transposon [176], recombining AAV [114], and CRISPR/Cas9 [121,128,169]. The main loci for transgene insertion in the creation of hypoimmunogenic cells are *AAVS1* [116,121,172] and *GAPDH* [128] (Table 3).

According to numerous data, insertion of the transgene into the *AAVS1* site located on chromosome 19 in intron 1 of *PPP1R12C* (encoding protein phosphatase 1 regulatory subunit 12C) ensures the stable expression of the transgene, and does not cause interference with *PPP1R12C*. However, there is evidence that DNA methylation suppresses transgene expression, for example, in hepatocytes and myeloid precursors derived from iPSCs [232]. In addition, it has been shown that transgene insertion into the *AAVS1* locus under the control of some promoters, such as *CMV7* [233] and *EF1α* [233,234], does not provide a sufficiently high level of expression. In addition, there is evidence that transgene insertion into this locus disrupts the *MBS85* gene [235]. Thus, the issue of the safety of using this locus in the clinic remains debatable.

Since *GAPDH* is constitutively expressed in all human cells, it was chosen as an attracting locus for transgene integration. If the transgene is integrated in the *AAVS1* locus without promoter control, the integration of the transgene into the *GAPDH* locus is performed through the insertion of the coding sequence of the transgene downstream of the last exon through 2A peptides [128,236]. Gerace et al. [128] performed *PDL1* insertion into *GAPDH* to prevent xeno-rejection, as previously Yoshihara et al. [127] showed that the overexpression of *PDL1* as a result of lentiviral transduction of cells prevents xeno-rejection. However, *PDL1* insertion into *GAPDH* did not block xeno-rejection [128]; lentiviral insertion was probably able to ensure the necessary level of expression, while the integration of *PDL1* into the *GAPDH* locus was not. Thus, the *GAPDH* locus is not a universal site for transgene integration.

In addition to these loci, *CCR5* and *hRosa26*, located on chromosome 3, are used as safe harbors in cellular genetic engineering. *CCR5* encodes the major co-receptor for HIV-1. A null mutation (∆32) in the *CCR5* gene in humans has been shown to result in repeated resistance to HIV-1-induced immunodeficiency [237]. *CCR5* is located in a gene-rich region surrounded by tumor-associated genes [238]. The *Rosa26* locus was discovered in mice via retroviral integration [239], and its ortholog in humans was subsequently found [238]. *hRosa26* is located in an intron of the gene encoding *THUMPD3,* the function of which has not yet been elucidated. In addition, proto-oncogenes are located near the *hRose* locus [238]. Thus, there are also doubts about the safety of using *CCR5* and *hRosa26* as a safe harbor in the context of clinical application.

Therefore, the main problem remains the choice of a “safe harbor” that satisfies a number of safety criteria. Aznauryan et al. [240], based on the criteria proposed earlier [238], proposed a number of the following criteria for choosing a “safe harbor” that ensures stable transgene expression and the greatest safety of its insertion.

50 kb away from known gene, to prevent genes located nearby from being affected;300 kb away from known oncogene, to prevent insertional oncogenesis;300 kb away from miRNAs, 150 kb away from lncRNAs and tRNAs, so as not to disrupt gene expression and cell cycle regulation;300 kb away from telomeres and centromeres, so as not to disrupt cell division;20 kb away from known enhancer region, so as not to interfere with enhancer–gene interaction.

Bioinformatics analysis allowed the authors to identify 2000 potential loci meeting these criteria, 5 of which were selected and screened. Thus, the authors found several sites that ensure long-term and stable transgene expression in vitro in stable and primary cell lines [240]. Using similar criteria, another “safe harbor” in T-cells has been identified and confirmed [241].

The main problem with using Cas9 or any other Cas nuclease to edit iPSCs is the high sensitivity of iPSCs to DSBs [242] via p53 activation [243]. Approaches with transient p53 inhibition led to the genomic instability of the resulting edited iPSC clones [244]. Therefore, safer advanced genomic editors should be used for therapeutic iPSC editing. In the following, we will review the capabilities of currently known CRISPR/Cas9-based advanced editors that work without generating DSBs.

#### 2.6.3. The Potential of Advanced CRISPR/Cas9 Tools for Engineering Immunoprivileged iPSC Lines

The insufficient safety of genomic Cas9 editors has prompted the development of alternative CRISPR/Cas9-based molecular tools for manipulating gene activity (epigenetic editors) or genomic editors operating on other molecular principles (prime editors). In the following, we will discuss their potential use for T1D therapy.

##### CRISPR/Cas9 Artificial Transcription Factors

In the following, we will consider the potential use of one group of epigenetic editors, called CRISPR/Cas9 artificial transcription factors, for T1D therapy. To turn the Cas9 nuclease into an artificial transcription factor, the nuclease activity of Cas9 is eliminated by D10A and H840A mutations, and the transcription activation or repression domains are fused to Cas9. The mutated Cas9 (called dead-nuclease Cas9, dCas9) is able to bind gRNA and search for a genomic target and bind it, while the regulatory domains require appropriate factors that either stimulate (in the case of transcription activation domains) or repress (in the case of repression domains) the activity of RNA polymerase, eventually functioning as an artificial activator or repressor of transcription [245].

The study of molecular mechanisms accompanying the differentiation of multipotent progenitor cells into pancreatic β-cells led to the discovery of a set of transcription factors responsible for this process [246,247,248]. The most active among these factors were found to be pancreatic and duodenal homeobox 1 (PDx1), protoendocrine factor (neurogenin 3, Ngn3), NK6 homeobox 1 (NKx6.1), and musculoaponeurotic fibrosarcoma oncogene family A (MaFA). In further experiments, the possibility of manipulating insulin production in HEK293 cells was demonstrated through the artificial activation of the *INS* insulin gene with CRISPR/dCas9-based transcription factors [249]. The stimulation of the insulin production has been achieved in pancreatic β-cell genes through the multiplex activation of *PDX1, NEUROG3, PAX4*, and *INS* genes using CRISPR/dCas9-VP160, CRISPR/dCas9-TET1 and CRISPR/dCas9-P300 epigenetic activators [250].

These results demonstrated the fundamental possibility of manipulating insulin metabolism in β-cells and raised the question of whether it is possible to activate insulin metabolism in heterologous, non-insulin-producing cells. Recently, CRISPR/dCas9-VPR delivered to HEK293 cells as ribonucleoprotein complexes on magnetic peptide Q-imprinted chitosan nanoparticles (MQIPs) and targeting *INS*, *PDX1*, *NGN3*, *NKX6.1*, and *MAFA* was shown to convert cells to glucose-dependent and insulin-producing cells [251]. The authors observed not only the induction of the expression of target transcription factors, but also of other proteins and signaling pathways that support insulin production. Moreover, the activation of insulin gene expression alone is sufficient to induce insulin production in HEK293 [252]. The activation of transcription factor genes *PDX1*, *NGN3*, *NKX6.1*, and *MAFA* is observed, but protein products are not detected. Interestingly, the activation of genes encoding proteins and signaling pathways that ensure insulin production is also observed. The existence of a small set of transcription factors that, when activated, reprogram cells into β-cells is reminiscent of the discovery of the Yamanaka cocktail, which revolutionized the technology of reprogramming of somatic cells into iPSCs [253].

However, Cas9-based artificial transcription factors have certain shortcomings. The currently found set of β-cell reprogramming factors must be continuously activated to maintain the key functions of reprogrammed cells. Thus, Cas9-based transcription factors must be continuously expressed, which raises questions about Cas9 delivery forms that could support prolonged Cas9 expression, as well as the enhanced immunogenicity of the persistent foreign Cas9 protein in cells. In addition, the number of off-target Cas9 binding sites is much larger than the number of off-target sites it can efficiently cut [177,254], so the number of undesirable changes in expression outside of target genes and genome fragments is potentially larger. In addition, the H840A mutation is insufficient to block the activity of the corresponding ruvC nuclease domain, and additional mutations in this domain are required for its complete inactivation [255].

##### Prime Editors

Despite the promising results of CRISPR/Cas9 artificial transcription factors, at least at the current stage of their development, it appears that they provide only short-term treatment associated with a number of off-target changes in gene expression. Therefore, an alternative genome editor is needed to create stable inherited genome changes resulting in gene knockouts and gene insertions necessary to create an immunoprivileged iPSC line. At the same time, such a genome editor should be safer, such in terms of as causing far less on- and off-target side effects compared to Cas9. Currently, prime editors are best suited for the role of the described advanced genome editor (PE).

The original PE is a chimeric protein complex that is a Cas9 nickase (H840A) fused to reverse transcriptase and an extended guide RNA (prime editing guide RNA, pegRNA) that both defines the target genome and carries a template encoding the desired genome edit [256]. Compared to Cas9, PE acts without generating DSBs and does not require exogenous donor DNA. Thus, PE generates far fewer by-products than Cas9 and, therefore, is much safer than Cas9. Since the original PE was created, a number of modifications have been made to the chimeric protein to improve its expression, reverse transcriptase activity, nuclear localization, chromatin accessibility, size, PAM preference, and genome editing results by inhibiting the competing mismatch DNA repair pathway, and to the pegRNA to improve its expression and stability, including the creation of a circularized pegRNA with remarkably enhanced activity (reviewed in [257]). Most of the earlier PE modifications allow only gene knockouts. However, a twinPE approach has recently been developed that, in combination with an improved site-specific Bxb1 integrase system, allows for fairly efficient (up to 6%) knockin of a large DNA fragment (about 6 kb) with a low level of undesired indels [258]. The presence of indels after primer editing can be explained by the insufficient inactivation of the HNH nuclease domain by the H840A mutation [255], and additional mutations should be introduced to make the HNH domain really “dead”.

To date, several examples of genome editing in iPSCs (e.g., [259,260,261]) and hESCs [262] are known. In some cases, the authors have had to use selection markers to increase the enrichment of edited cells, lentiviral or AAV vectors for PE delivery, and combinations of PE with other genetic engineering tools, such as its combination with the piggyBac transposon [261]. Of course, the use of antibiotic selection markers and lentiviral constructs is incompatible with therapeutic genome editing. Nevertheless, improved PE variants will drive progress in this field. In addition, specific approaches are being developed to enrich primary edited cells [263], which should facilitate the isolation of edited clones. Another weakness of PE is that part of the pegRNA scaffold sequence can sometimes be incorporated into the target genome [256]. This disadvantage may not be significant or even beneficial (if it causes a frameshift) in the case of PE-mediated gene knockouts. Although PE is not currently used for T1D therapy and is not currently being tested in gene knockouts and knockins associated with the production of hypoimmunogenic iPSCs, it clearly represents a safer alternative to Cas9 nuclease.

## 3. Conclusions

CRISPR/Cas-mediated genome editing and iPSC technology enable a new class of cell replacement therapies for the treatment of T1D. It is likely that their combination will accelerate the development of a β-cell replacement drug aimed at treating diabetes without the need for immunosuppression in a deviceless approach. However, the drawbacks of both cell technologies and CRISPR/Cas-editors must be carefully considered. β-cells obtained through differentiation or transdifferentiation protocols should be purified from immature β-like, non-targeted, or undifferentiated cells that may adversely affect β-cell activity. Allogeneic β-cells differentiated from genetically engineered low-immunogenic iPSCs are a cheap and rapid option for the treatment of T1D. However, their level of protection from the immune system is currently insufficient, resulting in significant levels of immune rejection in patients. The use of classical CRISPR/Cas system tools delivered in viral vectors can lead to the creation of off-target edits and insertional mutagenesis, which can lead to the cancerous transformation of low-immunogenic β-cells. This problem can be addressed by using high-fidelity Cas9 variants or advanced Cas9-based tools such as prime editors, delivering CRISPR/Cas systems as mRNA or RNA complexes, and carefully planning genome editing experiments to reduce the number of successive rounds of editing and cell clone selection to a minimum. To control the possible cancerous transformation of engineered β-cells, inducible non-immunogenic and cell state-independent suicide genes, such as inducible caspase 9, can be introduced. In addition, safe harbors for the introduction of protective genes should be carefully explored and selected so as not to disrupt the functions of surrounding genes. Perhaps our list of hidden problems and their solutions is not complete and further studies and clinical trials will uncover additional difficulties associated with hypoimmunogenic engineered β-cells. However, our current knowledge already allows us to foresee serious problems, the solution of which will greatly reduce the chance of turning engineered β-cells into a “Trojan horse” that will bring hidden problems to T1D patients after transplantation.

## Figures and Tables

**Figure 1 ijms-24-17320-f001:**
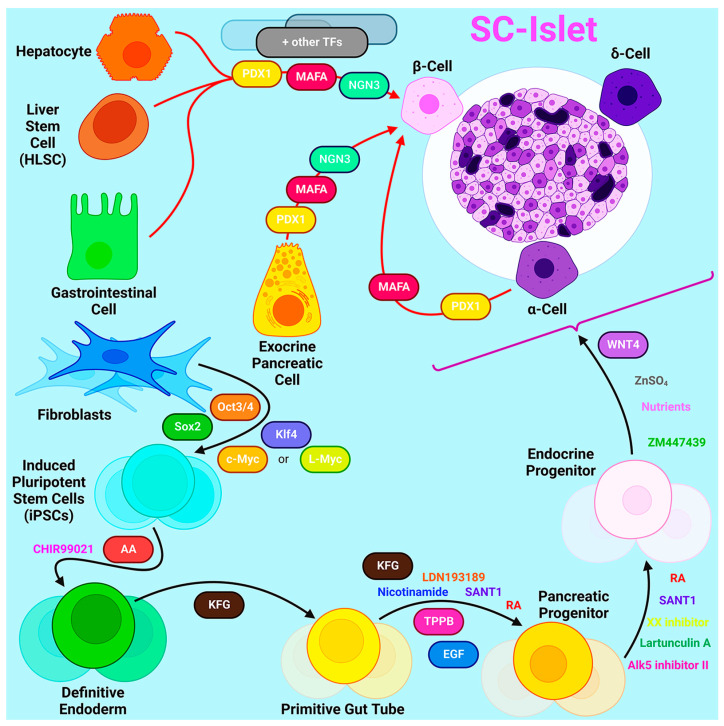
The ways to produce β-cell from other cell types by differentiation or transdifferentiation.

**Figure 2 ijms-24-17320-f002:**
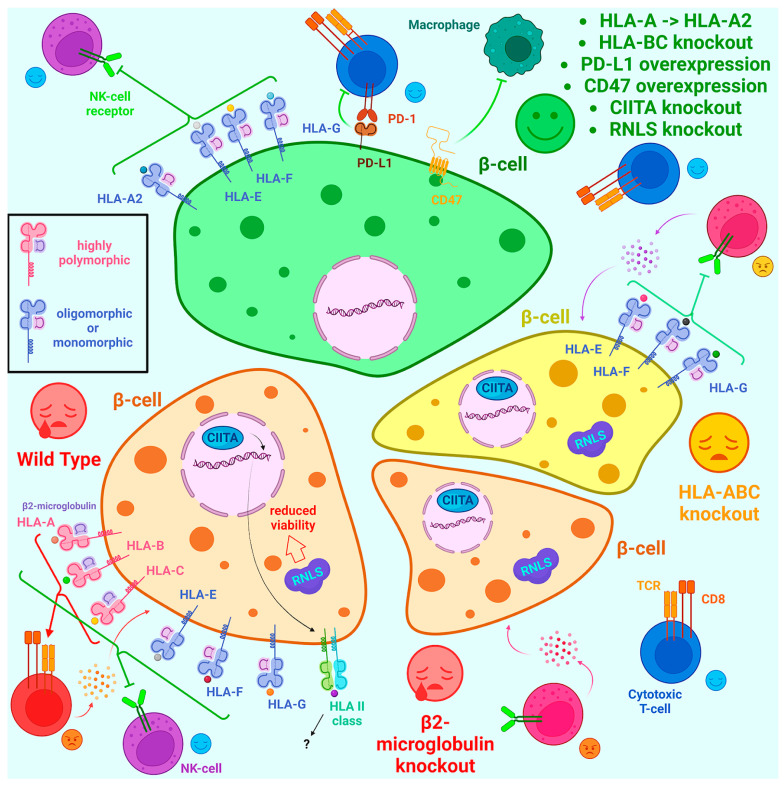
Steps to obtain low-immunogenic β-cells from corresponding engineered iPSCs.

**Figure 3 ijms-24-17320-f003:**
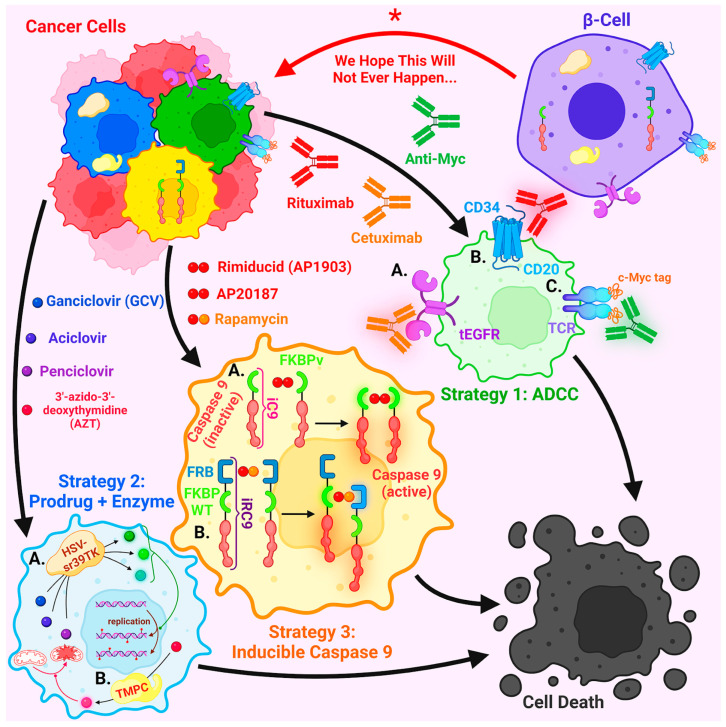
If iPSC-derived β-cells become cancerous, there are 3 possible strategies to kill them, adopted from CAR-T therapy. Strategy 1 involves the use of (**A**) cetuximab, rituximab, or anti-Myc monoclonal antibodies against antigens such as tEGFR, (**B**) CD20 or (**C**) Myc-tag on the cell surface, respectively, resulting in cell death via the ADCC mechanism. Strategy 2 is based on (**A**) the ability of HSV-TK to convert ganciclovir, acyclovir or penciclovir into a toxic product for the cell, resulting in cessation of replication, or (**B**) on the activity of TMPC, which phosphorylates AZT, resulting in loss of mitochondrial inner membrane potential. Strategy 3 involves caspase 9-induced dimerization; caspase9-FKBPv (iC9) homodimerizes upon addition of (**A**) rimiducid (or AP20187), and or caspase9-FKBP (WT)-FRB (iRC9) heterodimerizes upon addition of (**B**) rapamycin, respectively; both result in caspase 9 activation and induction of apoptosis. The asterisk indicates potential cancerous transformation of the genetically modified iPSC cell.

**Table 1 ijms-24-17320-t001:** Diabetes mellitus criteria (venous plasma).

Fasting Glucose Plasma (Fasting Is Defined as No Caloric Intake for at Least 8 h)	Random Plasma Glucose or 2-h Plasma Glucose during OGTT	Glycated Hemoglobin
≥7.0 mmol/L (126 mg/dL)	≥11.1 mmol/L (200 mg/dL).	≥6.5% (48 mmol/mol)

**Table 2 ijms-24-17320-t002:** Ongoing clinical trials of cell replacement therapies to treat patients with T1D.

Clinical Trial NCT	Company	Product	Cells	Administration
NCT03163511	Viacyte	VC-02	Pancreatic endoderm cells (PEC-01 cells)	Subcutaneous in a protective device
NCT03513939	Sernova	Cell Pouch	Therapeutic cells including islets	Abdominal musculature
NCT04786262	Vertex	VX-880	Allogeneic fully differentiated insulin-producing islets	Infused into the hepatic portal vein
NCT05565248	CRISPR Therapeutics AG in collaboration with Viacyte	VCTX211	Allogeneic pancreatic endoderm cells (PEC211) genetically modified using CRISPR/Cas9	In a surgically implanted durable, removable, perforated device
NCT05791201	Vertex	VX-264	Allogeneic fully differentiated insulin-producing islets	In a surgically implanted channel array protective device

**Table 3 ijms-24-17320-t003:** CRISPR/Cas9 genome editors used to generate hypoimmunogenic stem cells.

Type of Nuclease	Type of Delivery	Modification	Reference
spCas9n	Transfection plasmids	knockout *B2M* gene	[111]
spCas9	Nucleofection RNP	knockout *CIITA* gene	[113]
		knockout *HLA-B*, and *CIITA* genes	[125]
		knockout *HLA-B*, HLA-C, and *CIITA* genes	[124]
		knockout *B2M*, and *CIITA* genes	[123]
		knockout *HLA-A*, *HLA-B*, and *HLA-DR*	[168]
	Nucleofection plasmids	knockout *HLA-B*, *HLA-C*, and *CIITA* genes	[116]
		knockout *B2M*, knockin *HLA-G* in *B2M* locus	[169]
		knockout *HLA-B*, *HLA-C*, and *CIITA* genes	[121]
		knockin *PD-L1*, *HLA-G*, *CD47* in *AAVS1* locus	
		knockout *CIITA* gene	[122]
		knockout *HLA-B* gene	[170]
		knockout *B2M* gene	[171,172]
		knockout *HLA-B*, *HLA-C*, and *CIITA* genes	[116]
		knockin iC9 in *AAVS1*	
		knockout *B2M*, *CIITA*, and *PVR* genes	[173]
	Lipofection plasmids	knockout *B2M* gene	[112,174]
		knockout *B2M*, and *CIITA* genes	[120]
	Lipofection gRNA in cells with knockin inducible Cas9 in *AAVS1* locus	knockout *HLA-A*, *HLA-B*, *HLA-C*, and *CIITA* genes	[115]
	Lentiviral particles	knockout *B2M*, and *CIITA* genes	[175]
Hi-Fi spCas9	Nucleofection RNP	knockout *HLA-A*, *HLA-B*, and *HLA-C*	[176]
		knockout *B2M* gene	[128]
	Nucleofection plasmids	knockin *HLA-E*, *PD-L1*, *IL-2* mutein, *IL-10*, and *TGFB1* in *GAPDH* locus	[128]

## Data Availability

The datasets generated during and/or analyzed during the current study are available from the corresponding author on reasonable request.

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
