# Peer review of "Challenges of CRISPR/Cas-Based Cell Therapy for Type 1 Diabetes: How Not to Engineer a “Trojan Horse”"

_ijms, 2023, doi:10.3390/ijms242417320_

Round 1
Reviewer 1 Report
Comments and Suggestions for Authors
Karpov et al. comprehensively summarized the therapeutic approach for Type 1 diabetes, focusing on CRISPR-Cas9 cell engineering. It starts with broad information, including etiology and epidemiology, genetic predisposition and environmental triggers, diagnosis, treatment options and complications, and cell therapy. CRISPR/Cas9 approaches have been discussed in the cell therapy section, which covers the cell engineering of different cell types, including beta cells, immune cells, and PSC- or iPSc-derived insulin-producing cells. The review further discusses the issues and potential remedies for each problem. Although the lengthy and comprehensive summary of T1D is broad, it includes up-to-date clinically relevant information necessary to further discuss the option for cell therapy. Although this section is long, they have included sufficient information to discuss the options and challenges regarding CRISPR-Cas9 therapeutic approaches. I found this review article to be informative and easy to follow. I would recommend the review for publication.
Author Response
We appreciate your detailed review of our work. In response, we have modified the conclusion section to move away from generalities and provide a critical assessment of the issues raised in our review. Additionally, we have suggested possible solutionsReviewer 2 Report
Comments and Suggestions for Authors
Karpov et.al has written a nice review article. I have only one suggestion for the authors that if they could include a brief introduction paragraph for CRISPR/Cas system. This will be nice for the readers who are not familiar with the system and make the article complete. Further, If the authors do not want to do these changes it is also fine and could be accepted in the current form, but I will suggest the authors to add it. Overall, I Find this review article interesting and informative for the people in the field.
Comments on the Quality of English LanguageQuality of English is good.
Author Response
We appreciate of detailed review of our work. Therefore, we believe that adding more general information about the CRISPR/CAS system would make the work too extensive. Some general descriptions of the CRISPR/Cas9 system and its use for genome editing are provided in the introduction section (from line 57). In the revised manuscript we have modified sections 2.6.2 and have added more information about the advantages and disadvantages of using genetic editing, as well as new high-precision editors such as Prime Editors (paragraph 2.6.3.2)
Reviewer 3 Report
Comments and Suggestions for Authors
The manuscript provides an insightful analysis of the potential therapies for Type 1 Diabetes using cellular regenerative medicine approaches, particularly focusing on CRISPR/Cas-engineered cellular products. It is a timely and significant contribution to the field, given the urgent need for effective treatments for this condition. The manuscript offers a comprehensive overview of the current state of CRISPR/Cas-based cell therapy for Type 1 Diabetes, which is valuable for both newcomers to the field and established researchers. The discussion of potential drawbacks and the conceptualization of engineered cells as a "Trojan horse" is particularly thought-provoking and highlights the necessity for cautious optimism in the field. I have a few specific comments below:
Major:
• There is a notable omission of prime editing as a non-double strand break (DSB) generating alternative in the CRISPR/Cas9 toolkit. Including this recent advancement could provide a more holistic view of the genome editing landscape and its potential therapeutic applications.
• The risks associated with incorrectly engineered cell products are not clearly discussed. It would be beneficial for the manuscript to expand on the strategies for mitigating these risks and to provide a more detailed analysis of the potential consequences.
Comments on the Quality of English Language
There are several typographical and grammatical errors that need to be corrected. Below are two examples:
Line 108-109: “The average peak in global incidence ranges from 10 to 14 years [16], however in high risk populations it shifts towards 5-9 years”. It is confusing to read. The authors need to make it clear that it is referring to the years of age.
Line 200-201: “The regulation of carbohydrate metabolism at this period rests on the residual secretion of insulin by β-cells that usually appears for approximately 3 months” is confusing to read. The suggested alternative enhances clarity: "During this phase, the regulation of carbohydrate metabolism depends on the remaining insulin secretion by beta cells, which typically continues for about three months."
Author Response
The manuscript provides an insightful analysis of the potential therapies for Type 1 Diabetes using cellular regenerative medicine approaches, particularly focusing on CRISPR/Cas-engineered cellular products. It is a timely and significant contribution to the field, given the urgent need for effective treatments for this condition. The manuscript offers a comprehensive overview of the current state of CRISPR/Cas-based cell therapy for Type 1 Diabetes, which is valuable for both newcomers to the field and established researchers. The discussion of potential drawbacks and the conceptualization of engineered cells as a "Trojan horse" is particularly thought-provoking and highlights the necessity for cautious optimism in the field. I have a few specific comments below:
We appreciate your comprehensive review of our work. Please find the detailed responses to each comment below.
Major:
- There is a notable omission of prime editing as a non-double strand break (DSB) generating alternative in the CRISPR/Cas9 toolkit. Including this recent advancement could provide a more holistic view of the genome editing landscape and its potential therapeutic applications.
We have added paragraph 2.6.3.2. Prime editors
- The risks associated with incorrectly engineered cell products are not clearly discussed. It would be beneficial for the manuscript to expand on the strategies for mitigating these risks and to provide a more detailed analysis of the potential consequences.
We have made changes to the genome editing section (paragraph 2.6.2) by adding information about the potential risks associated with off-target, strategies to control Cas-9-induced off-targets and etc. Simultaneously, we have modified the conclusion section to move away from generalities and provide a critical assessment of the issues raised in our review. We have also suggested possible solutions&
Comments on the Quality of English Language
There are several typographical and grammatical errors that need to be corrected. Below are two examples:
Based on your comments about the level of English, we made several edits to the entire manuscript. It addition, it was reviewed by a colleague who is fluent in writing English.
Line 108-109: “The average peak in global incidence ranges from 10 to 14 years [16], however in high risk populations it shifts towards 5-9 years”. It is confusing to read. The authors need to make it clear that it is referring to the years of age.
The sentence has been corrected.
Line 200-201: “The regulation of carbohydrate metabolism at this period rests on the residual secretion of insulin by β-cells that usually appears for approximately 3 months” is confusing to read. The suggested alternative enhances clarity: "During this phase, the regulation of carbohydrate metabolism depends on the remaining insulin secretion by beta cells, which typically continues for about three months."
Thank you very much for your comment. We have replaced the confusing sentence.